# Health-Related Quality of Life with Six Domains: A Comparison of Healthcare Providers without Chronic Diseases and Participants with Chronic Diseases

**DOI:** 10.3390/jcm13185398

**Published:** 2024-09-12

**Authors:** Mohamad Adam Bujang, Yoon Khee Hon, Wei Hong Lai, Eileen Pin Pin Yap, Xun Ting Tiong, Selvasingam Ratnasingam, Alex Ren Jye Kim, Masliyana Husin, Yvonne Yih Huan Jee, Nurul Fatma Diyana Ahmad, Cheng Hoon Chew, Clare Hui Hong Tan, Sing Yee Khoo, Fazalena Johari, Alan Yean Yip Fong

**Affiliations:** 1Clinical Research Centre, National Institutes of Health, Sarawak General Hospital, Ministry of Health Malaysia, Kuching 93586, Sarawak, Malaysia; adam@crc.gov.my (M.A.B.); laiweihong@crc.moh.gov.my (W.H.L.); eileenyap.crc@gmail.com (E.P.P.Y.); tiongxt.crc@gmail.com (X.T.T.); khoosingyee@gmail.com (S.Y.K.); zazy2182@yahoo.com (F.J.); alanfong@crc.gov.my (A.Y.Y.F.); 2Institute for Clinical Research, Ministry of Health Malaysia, Block B4, National Institutes of Health (NIH), No. 1, Jalan Setia Murni U13/52, Seksyen U13, Shah Alam 40170, Selangor, Malaysia; masliyana.h@moh.gov.my; 3Department of Psychiatry and Mental Health, Sarawak General Hospital, Ministry of Health Malaysia, Kuching 93586, Sarawak, Malaysia; selvasingam@gmail.com; 4Quality Unit, Sarawak General Hospital, Ministry of Health Malaysia, Kuching 93586, Sarawak, Malaysia; alexkimsgh@gmail.com; 5Radiotherapy and Oncology Unit, Sarawak General Hospital, Ministry of Health Malaysia, Kuching 93586, Sarawak, Malaysia; yvonnejee91@hotmail.com; 6Heart Center, Sarawak General Hospital, Ministry of Health Malaysia, Kuching-Samarahan Expressway, Kota Samarahan 94300, Sarawak, Malaysia; nurulfatmadiyana86@gmail.com; 7Institute for Medical Research, Ministry of Health Malaysia, Block C, National Institutes of Health (NIH), No. 1, Jalan Setia Murni U13/52, Seksyen U13, Shah Alam 40170, Selangor, Malaysia; chewch@moh.gov.my; 8Division of Nephrology, Department of Medicine, Sarawak General Hospital, Ministry of Health Malaysia, Kuching 93586, Sarawak, Malaysia; clare.tan.hui.hong@gmail.com

**Keywords:** HRQ-6D, health-related quality of life

## Abstract

**Background/Objectives:** This study aims to compare the health-related quality of life (HRQOL) between healthcare providers without chronic diseases and participants with chronic diseases presenting with one of the four different primary diagnoses on the health-related quality of life with six domains (HRQ-6D) scale. **Methods:** This is a cross-sectional study to compare the HRQOL between healthcare providers without chronic diseases and participants with chronic diseases. Data collection was performed from May 2022 to May 2023. Data for the comparison group were taken from healthcare providers without chronic diseases, and for the participant group with chronic diseases, the data were collected from actual patients with one of four types of primary diagnoses who were recruited from specialist cardiology, oncology, psychiatry, and nephrology clinics. All the participants of this study filled in the HRQ-6D. **Results:** There were 238 (58.6%) healthcare providers without chronic diseases who participated in this study, as well as 41 (10.1%) patients with end-stage renal disease (ESRD), 48 (11.8%) patients with cancer, and 40 (9.9%) patients who were depressed, and the remaining patients had heart disease. The means (SD) of HRQ-6D scores among healthcare providers without chronic diseases for pain, physical strength, emotion, mobility, self-care, perception of future health, and overall HRQ-6D score were 75.3% (19.8), 74.5% (21.1), 85.6% (18.4%), 93.0% (12.3), 91.6% (13.9), 74.2% (23.3), and 82.4% (13.6), respectively. In comparisons between healthcare providers without chronic diseases and participants with chronic diseases, all mean differences of the overall HRQ-6D score and its domains and dimensions were statistically significant (*p* < 0.001). **Conclusions:** The overall score of the HRQ-6D, as well as its domains and dimensions are sensitive in detecting the study participants with chronic diseases from among those without chronic diseases. Therefore, the HRQ-6D is a reliable and valid scale to measure HRQOL. Future studies may use this scale for interventional, observational, and cost-effectiveness studies.

## 1. Introduction

Standardized and validated outcome measures are necessary for monitoring a disease process and evaluating the outcome of an intervention [1,2,3,4]. Apart from the conventional biochemical or clinical tests that serve as objective diagnostic tools for monitoring a patient’s current medical condition, patient-reported outcomes are equally important for assessing pain and overall quality of life [5,6,7]. This explains why health-related quality of life (HRQOL) has increasingly been recognized and used as an outcome measure for the effects of (chronic) medical conditions and their treatments on the daily functioning of life.

The health-related quality of life with six domains (HRQ-6D) scale was developed in 2023 and is reliable with excellent validity based on statistical measures and clinical evidence [8]. The HRQ-6D has 12 items, with two items allocated to each domain. These domains can be further categorized into the following three dimensions: “Health”, “Body function”, and “Perception”. Previously, clinical evidence was derived from the association of the HRQ-6D with healthcare workers who present with various health conditions [8]. So far, no evaluation has been made by using the HRQ-6D on actual patients with various confirmed diagnoses. This led to a proposed idea to further enhance the clinical utility of the HRQ-6D by testing it on a group of patients and then comparing the HRQ-6D scores between participants with chronic diseases and those without chronic diseases.

To achieve this goal, this study is designed to assess the HRQOLs of participants with chronic diseases and those without chronic diseases and to compare them by using the HRQ-6D. Those with chronic diseases are represented by diverse groups of patients with one of four primary diagnoses (i.e., end-stage renal disease (ESRD), cancer, depressive disorders, and heart disease) who were recruited in this study. This is to determine whether the HRQ-6D and its domains and dimensions are sensitive in detecting differences between participants with chronic diseases and those without chronic diseases and to subsequently prove that the HRQ-6D is a reliable and valid scale for measuring HRQOL.

## 2. Materials and Methods

This cross-sectional study aimed to compare the HRQOL using the HRQ-6D to detect any difference(s) in HRQOL between participants with chronic diseases and those without chronic diseases. Data collection was performed during the period from May 2022 to May 2023. For participants without chronic diseases, the study sample was recruited from healthcare workers working in Sarawak General Hospital and The Heart Centre, Sarawak General Hospital. Both are governmental healthcare facilities under the administration of the Ministry of Health, Malaysia. The selection criteria for this group included (i) workers who are currently working in a healthcare setting and are (ii) 18 years old and above, (iii) without chronic diseases and not currently taking any medications (based on self-reported responses), and (iv) agree to participate in this study. 

For the group of participants with chronic diseases, the data were collected from actual patients who presented with one of four types of primary diagnoses and were recruited from specialist cardiology, oncology, psychiatry, and nephrology clinics. All the participants of this study filled in the HRQ-6D. The selection criteria for the participants with chronic diseases included (i) patients who are currently under follow-up in four specialist clinics who were recruited from cardiology (and heart disease) or oncology (any type of cancer), patients with depressive disorders who were recruited from psychiatry, and end-stage renal disease (ESRD) patients from nephrology specialist clinics who are (ii) 18 years old and above and (iii) agree to participate in this study. However, study respondents who were unconscious, too sick, or in a comatose state and those who had an unstable mental condition during the recruitment period were excluded from this study. 

The recruitment of all study participants for this study was conducted in each of the respective specialist clinics. Most of the patients were in a stable condition, as any patients who presented with too severe conditions were excluded from this study. The selection was based on consecutive sampling and voluntary participation. 

### 2.1. Ethical and Regulatory Considerations

Only study participants who had given informed consent for participating in this study were surveyed. This study adhered to all the relevant guidelines and regulations stipulated by the Medical Research and Ethics Committee (MREC), National Institutes of Health, Ministry of Health Malaysia. Ethical approval for this study was granted on 22-November-2022 [Tuesday] by the Medical Research and Ethics Committee (MREC), NMRR ID-21-01979-XDL (IIR).

### 2.2. HRQ-6D Scale

The HRQ-6D scale measures six domains, each represented by two items. The full set of HRQ-6D items and the scoring mechanism are presented in a previous paper published elsewhere [8]. The HRQ-6D domains consist of pain, physical energy, emotional symptoms, mobility, self-care, and perception of future health. These six domains can be further categorized into the following three dimensions: health, body function, and perception [8]. The overall and domain scores of HRQ-6D are presented as percentages, with higher scores indicating better HRQOL.

### 2.3. Sample Size Planning

The sample size statement for this study was based on a guideline for sample size determination that was introduced in a previous study [9]. This study aims to measure the HRQOL in two different groups of participants by using the HRQ-6D. The objective of this study necessitates a multivariate analysis, such as an analysis of covariance (ANCOVA), since the analysis necessitates any covariates or potential confounders to be adjusted for in the statistical analysis. Hence, an easier approach to determining the sample size requirement for this study is based on a rule of thumb for sample size determination of the general linear model ANCOVA. Based on this recommendation, the minimum sample size of 300 participants is deemed sufficient for deriving accurate estimates for conducting ANCOVA in the target population [10]. By incorporating an additional 10.0% to make allowances for the possibility of non-response, the sample size for this study requires a minimum of 334 participants. 

### 2.4. Statistical Analysis

A descriptive analysis was used to describe the profile and the comparisons of HRQ-6D among five different groups of study respondents. For the univariate analysis, a one-way analysis of variance (ANOVA) was applied. Then, a multivariate analysis based on a general linear model ANCOVA was applied to compare the HRQ-6D scores among all the different groups, with an adjustment for gender and age. Subsequently, the Bonferroni procedure was applied as a post hoc test to determine which specific group differences are statistically significant by performing multiple comparisons of group means in a pairwise fashion. It is one of the statistical tests that can be applied when the results obtained from Levene’s test are not significant. All the analyses were conducted by using SPSS (IBM Corp. Released 2011. IBM SPSS Statistics for Windows, Version 20.0. Armonk, NY, USA: IBM Corp).

## 3. Results

The study population consisted of 238 (58.6%) healthcare providers without chronic diseases, 41 (10.1%) patients with ESRD, 48 (11.8%) patients with cancer, and 40 (9.9%) patients with depressive disorders, and the remaining patients had heart problems. The majority of the participants were female (71.5%), between 18 and 35 years old (46.7%), and Malay (36.5%) (Table 1).

### 3.1. HRQ-6D among Healthcare Providers without Chronic Diseases versus Participants with Chronic Diseases

The sample estimate means (SD) of HRQ-6D among healthcare providers without chronic diseases for pain, physical strength, emotion, mobility, self-care, perception of future health, and overall HRQ-6D score were 75.3% (19.8), 74.5% (21.1), 85.6% (18.4%), 93.0% (12.3), 91.6% (13.9), 74.2% (23.3), and 82.4% (13.6), respectively. In a comparison of HRQOL between participants with chronic diseases and healthcare providers without chronic diseases, all mean differences of overall HRQ-6D, together with all its domains, and dimensions were found to be statistically significant (*p* < 0.001) (Table 2 and Table 3).

### 3.2. Results Based on the Post Hoc Bonferroni Multiple Comparisons Test by Domains

The results are presented in Table 2. Based on the three domains of mobility, perception of future health, and overall HRQOL, all patients with each of the four different primary diagnoses reported poorer HRQOL in comparison to those from the group of healthcare providers without chronic diseases (*p* < 0.001). For the domain of self-care, nearly all patients with any of the primary diagnoses reported poorer HRQOL, except for cancer patients (*p* < 0.001). Nevertheless, the score of self-care for cancer patients is still lower than the group without chronic diseases. For the other three domains, including pain, physical strength, and emotion, only the depressive group reported poorer HRQOL compared to the group without chronic diseases. Moreover, the depressive group also reported poorer emotion when compared with patients who presented with any of the other three primary diagnoses, as follows: end-stage renal disease (ESRD), heart disease, and cancer. 

### 3.3. Results Based on the Post Hoc Bonferroni Multiple Comparisons Test by Dimensions

The results are presented in Table 3. Based on the two dimensions of body function and perception of future health, all patients with any of the different primary diagnoses reported poorer HRQOL in comparison to that of the group of healthcare providers without chronic diseases (*p* < 0.001). For the dimension of health, the depressive group reported poorer health compared with any other groups (*p* < 0.001). However, all patients who presented with any one of the four primary diagnoses reported lower scores in the health dimension when compared to that of the group without chronic diseases. For the other domains, such as pain, physical strength, and emotion, only the depressive group reported poorer HRQOL as compared to the group without chronic diseases. The depressive group also reported poorer emotion as compared to all other groups of patients who presented with any one of the three primary diagnoses, as follows: end-stage renal disease (ESRD), heart disease, and cancer.

## 4. Discussion

This study successfully demonstrated the criterion validity of the HRQ-6D based on a known-groups comparison, that is, the ability of the domains and dimensions scores to discriminate between well and ill groups of respondents (with a “healthy” condition and a “disease” condition, respectively). The analysis has been tested in both healthcare providers without chronic diseases and in actual patients with one of four primary diagnoses of chronic diseases, namely, end-stage renal disease (ESRD), cancer, depressive disorders, and heart disease, within the study population.

The HRQ-6D is designed to accurately measure the current status of HRQOL, which can be used for routine clinical practice (i.e., during an assessment of patients’ current health outcome(s)) and in clinical research (i.e., during a measurement of the clinical effectiveness of an intervention). The development of the HRQ-6D is based on the premise that an overall score for the HRQ-6D should aptly be utilized to represent the overall health HRQOL and all the three dimensions shall collectively represent the HRQOL in terms of an individual’s health, body function, and perception. These three dimensions can be further divided into six individual domains, which collectively provide a more specific measurement of HRQOL, thus serving as a valid tool within clinical and research contexts. All these dimensions are supported by the existing literature, and in this study, they were derived from the full extent of the overall “quality-of-life” concept introduced by the previous HRQOL scales [11,12,13,14,15,16].

All these findings indicate that cumulative clinical evidence points to the clinical relevance of the HRQ-6D. Indeed, the results of this study have indicated that patients with depressive symptoms reported lower HRQOL scores by using the HRQ-6D, suggesting that they experience poorer HRQOL compared to patients with other chronic diseases, even when they were in stable conditions. This study, therefore, demonstrates that clinical evidence does correlate with the HRQ-6D, which deems it to be a suitable generic instrument for measuring the HRQOL of both healthy individuals without chronic diseases and actual patients. The generic nature of the HRQ-6D also means that it is important to assess the various medical conditions in which the measures perform well. Future research shall therefore be directed toward the goal of improving the applicability of HRQ-6D as a generic instrument to all persons, irrespective of their type or number of illnesses. This is because it is likely that the HRQ-6D may still not be sensitive to some problems that are unique to particular diseases since HRQOL is a multidimensional concept, and different aspects of quality need different methods of measurement [17,18].

Due to the plethora of HRQOL instruments available nowadays, cumulative evidence comparing the performance of each of these instruments is constantly required to inform the selection of the most appropriate instrument for a wide range of applications, and such evidence also requires cumulative results from different settings and types of study [19,20]. Therefore, the aim of conducting future studies is to investigate the responsiveness of the HRQ-6D within a larger sample of people with a wide range of varying medical conditions and disease states and to compare the HRQOL measures that are elicited by the HRQ-6D among many differing medical conditions. This will support the future use of the measures by providing further evidence regarding their psychometric performance. 

### 4.1. HRQ-6D Cut-Off Scores for the HRQOL Category

Based on a Likert scale of five, the higher the score (i.e., in closer proximity to “strongly agree”), the higher the magnitude of a respondent to experience a lower quality of life (i.e., more pain, weaker physical strength, etc.). Therefore, those who rated the HRQ-6D items with a Likert scale score of one or two will be more likely to experience a better HRQOL. Hence, the classification of different categories of HRQOL based on their HRQ-6D scores is presented in Table 4.

The recommended cut-off scores of HRQ-6D are as follows: more than 80.0% is considered “healthy”, between 70 and 80 is considered “moderately healthy”, between 50 and 69 is considered “moderately poor health”, and less than 50.0% is considered “poor health”. For a respondent to be judged as “healthy” by HRQ-6D, he/she must score at least 8 out of 10 points (as there are two items within one domain and each has a Likert scale of five). Thus, the minimum score of eight points can be derived from either [i] both items being scored as four points or [ii] one item being scored as five points or the other item being scored as three points. This also means that a respondent is considered to be in “poor health” if he/she has obtained a score below average (since “below average” for 10 points refers to anything less than 5 points). The cut-off score for “moderate health” will be between 70 and 80 points.

The mean (95%CI) scores for the HRQ-6D domains and dimensions among healthcare providers without chronic diseases are used for comparison purposes. These participants do not suffer from chronic diseases, and they are categorized as “healthy” (scores > 80.0% in all domains and dimensions), except for the domains of pain, physical strength, and perception of future health. The rationale of this is that they are actually healthcare workers, and the majority of them possibly filled in the answers for this questionnaire during or after working hours when some of them might have felt tired. This also explains why they did not score above 80.0% in the two domains of pain and physical strength.

Although all of them are considered “healthy”, it is still logical for some of them to have thought that their health might deteriorate within the next five years due to aging. This explains why their score regarding the perception of future health is less than 80.0%. However, their scores are much higher compared to participants who have already experienced some chronic diseases. The authors postulate that a young and healthy person in the early morning before engaging in any intensive physical activity may easily score more than 80.0% or even 90.0% in all the domains of HRQ-6D.

The idea to develop the cut-off scores of the HRQ-6D is to increase the ease of interpretation. Some studies might be interested in comparing the different categories of health statuses instead of merely reporting the mean scores. In addition, it is also possible for some researchers to prefer to report the HRQOL outcome measure as a categorical or binary variable so that a comparison between different categories of health status (such as “healthy”, “moderately healthy”, “moderately poor health”, and “poor health”) can be made more simply using logistic regression.

### 4.2. Limitations of the Study

All patients are in stable conditions, despite having been afflicted by severe chronic diseases such as cancer and ESRD. Therefore, this study might not be able to elicit responses from patients who are in less stable conditions or whose conditions are deteriorating rapidly. All patients were invited to participate in this study based on their ability to adhere to strict eligibility criteria for this study, without having to pay particular attention to their specific disease conditions. The selection was based on consecutive sampling and voluntary participation, which has the disadvantage of being a non-probability sampling. Future studies may address this issue by conducting a survey using the same scale and focusing on each chronic disease individually with a larger sample size. Although the number of participants with chronic diseases is relatively small, the results were proven to be statistically significant, indicating that there is emphatically adequate statistical power to test the hypotheses. Finally, there is a potential limitation to the generalizability of the research findings due to the inclusion of only healthcare workers as the comparison group. However, this study aims to specifically select only healthcare workers for comparison because of two reasons, namely, [i] they are well-versed in a wide variety of disease conditions and so their self-assessment of health status is likely to be accurate and valid and [ii] they undergo health screenings more frequently, as required by the hospital authorities, compared to the general public.

## 5. Conclusions

This study successfully validated the HRQ-6D based on clinical evidence. The overall scores, three dimensions, and six domains are all sensitive in detecting those who have chronic diseases and those who do not have chronic diseases in various aspects. Therefore, the HRQ-6D is a reliable and valid scale to measure HRQOL.

A major limitation of this study is the inability to measure the true impact of HRQOL among chronic patients with severe conditions. Although the sample size is small, the results still demonstrated statistically significant findings, showing that the HRQ-6D is sensitive in differentiating between individuals with and without chronic diseases. However, future studies can apply the HRQ-6D to a broader range of populations, including patients with various diseases, and a larger sample size in order to obtain a better representation of HRQOL in these populations.

As this is the first study to use the HRQ-6D in a human population with chronic diseases, future studies may consider a head-to-head comparison of different HRQOL scales, specifically comparing the application of the HRQ-6D with other HRQOL measures. Furthermore, we shall also propose to conduct future studies that will utilize this scale for interventional studies (i.e., clinical trial studies), observational studies (i.e., measuring patient outcomes in short- and long-term cohort studies), and cost-effectiveness studies.

## Figures and Tables

**Table 1 jcm-13-05398-t001:** Basic demographic profile and primary diagnosis of subjects.

Profile	Category	*n*	%
Gender	Male	115	28.5
	Female	288	71.5
Age group	18–35	188	46.7
	36–40	75	18.6
	41–50	72	17.9
	51–60	45	11.2
	More than 60	21	5.2
	Not reported	2	0.5
Ethnic	Malay	147	36.5
	Iban	70	17.4
	Bidayuh	71	17.6
	Other Sarawakian	17	4.2
	Chinese	85	21.1
	Others	6	1.4
	Not reported	7	1.7
Primary diagnosis	Nil	238	59.1
	ESRD	40	9.9
	Cancer	48	11.9
	Depressive disorders	38	9.4
	Heart disease	39	9.7

**Table 2 jcm-13-05398-t002:** Comparison of HRQ-6D domains in patients with various primary diagnoses.

Domains		Mean	SD	LB	UB	Adj. Mean	LB	UB	Sig.
Pain	a	75.3	19.8	72.7	77.8	77.2	73.0	81.4	d
	b	68.5	19.2	62.4	74.6	69.6	63.0	76.1	
	c	66.1	20.0	59.8	72.4	67.1	60.7	73.4	
	d	58.9	22.4	51.6	66.3	61.0	53.6	68.4	a
	e	65.9	20.1	59.4	72.4	65.5	58.8	72.2	
Physical strength	a	74.5	21.1	71.8	77.1	77.4	73.0	81.7	d
	b	67.5	21.1	60.8	74.2	68.8	62.1	75.5	d
	c	66.6	18.0	60.9	72.3	66.2	59.7	72.7	d
	d	47.9	20.9	41.0	54.8	51.3	43.7	58.9	a, b, c, e
	e	66.4	18.8	60.3	72.5	65.4	58.5	72.2	d
Emotion	a	85.6	18.4	83.2	87.9	90.3	86.4	94.2	d
	b	78.0	20.9	71.3	84.7	80.0	74.1	85.9	d
	c	80.0	17.1	74.0	86.0	80.1	73.8	86.4	d
	d	43.8	16.0	38.4	49.1	49.3	42.5	56.1	a, b, c, e
	e	81.5	17.3	75.4	87.5	80.3	73.9	86.7	d
Mobility	a	93.0	12.3	91.5	94.6	93.1	90.0	96.3	b, c, d, e
	b	75.3	18.1	69.5	81.0	75.7	71.0	80.5	a
	c	80.0	14.3	75.2	84.8	80.3	75.5	85.2	a
	d	82.1	20.3	75.4	88.8	82.0	76.6	87.4	a
	e	75.1	16.6	69.6	80.7	75.7	70.8	80.7	a
Self-care	a	91.6	13.9	89.8	93.4	94.3	90.9	97.7	b, d, e
	b	76.0	20.6	69.4	82.6	77.4	72.2	82.6	a
	c	84.5	15.4	79.4	89.5	84.6	79.3	89.8	
	d	74.9	21.6	67.7	82.1	77.8	71.8	83.7	a
	e	77.1	17.1	71.5	82.7	76.4	71.0	81.9	a
Perception of future health	a	74.2	23.3	71.2	77.2	77.6	72.6	82.6	b, c, d, e
	b	55.0	22.9	47.7	62.3	56.9	49.3	64.6	a
	c	59.0	26.0	50.7	67.3	58.8	51.3	66.3	a
	d	55.0	25.9	46.5	63.5	58.4	49.7	67.1	a
	e	61.7	19.3	55.1	68.2	60.3	52.1	68.4	a
Overall HRQOL	a	82.4	13.6	80.6	84.1	85.5	82.5	88.6	b, c, d, e
	b	70.0	15.3	65.2	74.9	71.7	67.1	76.2	a
	c	71.7	14.5	66.4	76.9	71.5	66.5	76.5	a, d
	d	61.1	14.3	56.2	65.9	64.4	59.1	69.7	a, d, e
	e	71.0	13.9	66.0	76.1	70.1	65.0	75.2	a, d

Note: a: Primary diagnosis = Nil (healthy); b: primary diagnosis = end-stage renal disease; c: primary diagnosis = cancer; d: primary diagnosis = depressive disorders; e: primary diagnosis = heart disease; Adj. mean = adjusted mean; LB = lower bound; 95% confidence interval; UB = upper bound 95% confidence interval; Sig. = significantly different based on a pairwise comparison.

**Table 3 jcm-13-05398-t003:** Comparison of HRQ-6D dimensions in patients with various primary diagnoses.

Domains		Mean	SD	LB	UB	Adj. Mean	LB	UB	Sig.
Health	a	78.4	17.2	76.2	80.6	81.7	78.1	85.3	d
	b	71.3	17.6	65.7	77.0	72.7	67.2	78.2	d
	c	70.2	15.4	64.8	75.6	70.3	64.4	76.1	d
	d	50.5	17.2	44.7	56.2	54.1	47.8	60.4	a, b, c, e
	e	71.4	15.4	66.0	76.7	70.4	64.4	76.4	d
Body function	a	92.3	12.4	90.7	93.9	93.7	90.6	96.8	b, c, d, e
	b	75.6	18.3	69.8	81.5	76.6	71.9	81.2	a
	c	82.3	13.2	77.9	86.7	82.5	77.7	87.3	a
	d	78.5	19.4	72.1	85.0	79.9	74.5	85.2	a
	e	75.9	16.2	70.6	81.3	76.0	71.0	80.9	a
Perception of future health	a	74.2	23.3	71.2	77.2	77.6	72.6	82.6	b, c, d, e
	b	55.0	22.9	47.7	62.3	56.9	49.3	64.6	a
	c	59.0	26.0	50.7	67.3	58.8	51.3	66.3	a
	d	55.0	25.9	46.5	63.5	58.4	49.7	67.1	a
	e	61.7	19.3	55.1	68.2	60.3	52.1	68.4	a

Note: a: Primary diagnosis = Nil (healthy); b: primary diagnosis = end-stage renal disease; c: primary diagnosis = cancer; d: primary diagnosis = depressive disorders; e: primary diagnosis = heart disease; Adj. mean = adjusted mean; LB = lower bound; 95% confidence interval UB = upper bound 95% confidence interval; Sig. = significantly different based on a pairwise comparison.

**Table 4 jcm-13-05398-t004:** HRQOL classification based on HRQ-6D scores.

HRQ-6D	Healthy Subjects Mean (95%CI)	Healthy	Moderately Healthy	Moderately Poor	Poor Health
*Domains*					
Pain	77.2 (73.0, 81.4)	>80	70 ≤ 80	50 ≤ 69	<50
Physical strength	77.4 (73.0, 81.7)	>80	70 ≤ 80	50 ≤ 69	<50
Emotion	90.3 (86.4, 94.2)	>80	70 ≤ 80	50 ≤ 69	<50
Mobility	93.1 (90.0, 96.3)	>80	70 ≤ 80	50 ≤ 69	<50
Self-care	94.3 (90.9, 97.7)	>80	70 ≤ 80	50 ≤ 69	<50
Perception	77.6 (72.6, 82.6)	>80	70 ≤ 80	50 ≤ 69	<50
Overall	85.5 (82.5, 88.6)	>80	70 ≤ 80	50 ≤ 69	<50
*Dimensions*					
Health	81.7 (78.1, 85.3)	>80	70 ≤ 80	50 ≤ 69	<50
Body function	93.7 (90.6, 96.8)	>80	70 ≤ 80	50 ≤ 69	<50
Perception	77.6 (72.6, 82.6)	>80	70 ≤ 80	50 ≤ 69	<50

Note: Data from healthy subjects are collected for this study; these subjects are without any reported co-morbidities and/or chronic diseases.

## Data Availability

The data presented in this study are available on request from the corresponding author. Formal data requests must be made to the Director General Ministry of Health Malaysia. Once granted approval, the anonymous data will be made available.

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
