# Peer review of "Health-Related Quality of Life with Six Domains: A Comparison of Healthcare Providers without Chronic Diseases and Participants with Chronic Diseases"

_jcm, 2024, doi:10.3390/jcm13185398_

Round 1
Reviewer 1 Report
Comments and Suggestions for Authors
The study compares health-related quality of life (HRQOL) using the HRQ-6D instrument between healthy participants and those with health conditions, categorized into four primary diagnoses: end-stage renal disease (ESRD), cancer, depressive disorders, and heart disease. The finding validates the HRQ-6D based on clinical evidence.
I have the following questions and suggestions:
1. The terminology "healthy" versus "unhealthy" participants requires careful reconsideration. Health is a complex and multidimensional concept. The World Health Organization (WHO) defines health as the well-being of physical, mental, and social states. However, the criteria for a "healthy" population in this study—"without chronic disease and not self-reporting taking medicine"—does not necessarily imply true health. It might be more accurate to describe the groups as populations "with" and "without chronic diseases."
2. The healthy group consists of healthcare workers. Could this choice of population impact the representativeness of your comparisons? Please elaborate on this issue.
3. The results support the use of HRQOL in clinical settings to measure quality of life. How does HRQOL compare to other quality of life measures currently used in clinical settings?
4. Participants diagnosed with depression show significantly lower scores across various domains and dimensions compared to those with other diagnoses. Does this suggest that HRQOL is particularly sensitive to the quality of life impacts of depressive disorders? What implications might this have for the use of HRQOL in clinical settings?
5. How do HRQOL responses vary by gender and age? Is there a significant pattern in the variation of responses across these demographics?
Author Response
For research article
Health-Related Quality of Life with Six Domains (HRQ-6D): A Comparison of Participants Without and With Chronic Diseases
[Manuscript ID: jcm-3133929]
|
Response to Comments from Reviewer 1 Author's Reply to the Review Report (Reviewer 1) |
||
|
1. Summary |
|
|
|
Thank you very much for taking the time to review this manuscript. Please find the detailed responses below and the corresponding revisions/corrections highlighted/in track changes in the re-submitted files.
|
||
|
2. Questions for General Evaluation |
Reviewer’s Evaluation |
Response and Revisions |
|
Does the introduction provide sufficient background and include all relevant references? |
Can be improved |
|
|
Are all the cited references relevant to the research? |
|
|
|
Is the research design appropriate? |
Yes |
|
|
Are the methods adequately described? |
Yes |
|
|
Are the results clearly presented? |
Yes |
|
|
Are the conclusions supported by the results? |
Can be improved |
|
|
3. Point-by-point response to Comments and Suggestions for Authors |
||
|
Comments 1: The terminology "healthy" versus "unhealthy" participants requires careful reconsideration. Health is a complex and multidimensional concept. The World Health Organization (WHO) defines health as the well-being of physical, mental, and social states. However, the criteria for a "healthy" population in this study—"without chronic disease and not self-reporting taking medicine"—does not necessarily imply true health. It might be more accurate to describe the groups as populations "with" and "without chronic diseases."
|
||
|
Response 1: Thank you for pointing this out. We agree with this comment. Therefore, we have followed the reviewer’s suggestion by replacing the word ‘healthy’ with the phrase ‘without chronic diseases’ and the word ‘unhealthy’ (or ‘non-healthy’) with the phrase “with chronic diseases” in both the main text of this manuscript, as well as in its four tables.
|
||
|
Comments 2: The healthy group consists of healthcare workers. Could this choice of population impact the representativeness of your comparisons? Please elaborate on this issue. |
||
|
Response 2: Agree. We have, accordingly, made the necessary revision by inserting the following paragraph in the section on the limitations of this study. “There is a potential limitation to the generalizability of the research findings due to the inclusion of only healthcare workers as the comparison group. However, this study aims to specifically select only healthcare workers for comparison because of following two rea-sons, namely: [i] they are well-versed in a wide variety of disease conditions and so their self-assessment of health status is likely to be accurate and valid, [ii] they undergo health screenings more frequently as required by the hospital authorities, compared to the general public.”
Comments 3: The results support the use of HRQOL in clinical settings to measure quality of life. How does HRQOL compare to other quality of life measures currently used in clinical settings?
Response 3: Agree. We have, accordingly, made the necessary revision by inserting the following sentences in the conclusion of this study. “This is the first study to use the HRQ-6D in a human population with chronic diseases. Future studies may consider a head-to-head comparison of different HRQOL scales, specifically comparing the application of the HRQ-6D with other HRQOL measures.”
Comments 4: Participants diagnosed with depression show significantly lower scores across various domains and dimensions compared to those with other diagnoses. Does this suggest that HRQOL is particularly sensitive to the quality of life impacts of depressive disorders? What implications might this have for the use of HRQOL in clinical settings?
Response 4: Agree. We have, accordingly, made the necessary revision by inserting the following sentences within the discussion section of this research article. “Indeed, the results of this study have indicated that patients with depressive symptoms reported lower HRQOL scores simply suggest that they experience poorer HRQOL com-pared to patients with other chronic diseases, even when in a stable condition.”
Comments 5: How do HRQOL responses vary by gender and age? Is there a significant pattern in the variation of responses across these demographics? Response 5: Agree. We have, accordingly, accommodated the necessary requirement by applying ANCOVA in the proposed analysis, which is a multivariate technique commonly employed to adjust for the effects of confounders such as age and gender. The results show that ANCOVA has effectively eliminated the confounding effect of age and gender on HRQOL scores, by ensuring that there is no significant impact of age and/or gender on the outcome [i.e. HRQOL scores]. Hence, the results shall be able to indicate that the overall HRQOL and its domains are primarily affected by the disease condition.
|
||
|
4. Response to Comments on the Quality of English Language |
||
|
Point 1: No comment. |
||
|
|
||
|
5. Additional clarifications |
||
|
None. |
||

Reviewer 2 Report
Comments and Suggestions for Authors
The manuscript is well-organized and presents a thorough validation of the HRQ-6D scale. The clear structure and logical flow of the paper make it easy to follow, and the methods and results are presented in a manner that enhances understanding. The study contributes meaningfully to the field of health-related quality of life (HRQOL) measurement, particularly by providing a validation of the HRQ-6D scale in a diverse patient population. This is an important step in establishing the scale's utility in both clinical and research settings.
While the inclusion criteria are clearly stated, it might be helpful to provide more detail on how participants were recruited, particularly for the patient groups. For example, was there any consideration of disease severity or duration in the selection process?
While the discussion mentions the study's limitations, expanding on how these limitations might affect the interpretation of the results would be beneficial. For example, how might the relatively small sample sizes for some patient groups influence the findings?
Author Response
For research article
Health-Related Quality of Life with Six Domains (HRQ-6D): A Comparison of Participants Without and With Chronic Diseases
[Manuscript ID: jcm-3133929]
|
Response to Comments from Reviewer 2
|
||
|
1. Summary |
|
|
|
Thank you very much for taking the time to review this manuscript. Please find the detailed responses below and the corresponding revisions/corrections highlighted/in track changes in the re-submitted files.
|
||
|
2. Questions for General Evaluation |
Reviewer’s Evaluation |
Response and Revisions |
|
Does the introduction provide sufficient background and include all relevant references? |
Yes |
|
|
Are all the cited references relevant to the research? |
Yes |
|
|
Is the research design appropriate? |
Yes |
|
|
Are the methods adequately described? |
Yes |
|
|
Are the results clearly presented? |
Yes |
|
|
Are the conclusions supported by the results? |
Yes |
|
|
3. Point-by-point response to Comments and Suggestions for Authors |
||
|
Comments 1: While the inclusion criteria are clearly stated, it might be helpful to provide more detail on how participants were recruited, particularly for the patient groups. For example, was there any consideration of disease severity or duration in the selection process?
|
||
|
Response 1: Thank you for pointing this out. We agree with this comment. Therefore, we have mentioned in the methods section of this manuscript that “Recruitment was conducted in the respective four different specialist clinics. Most of the patients were in stable condition, as we had specifically mentioned in the methods section that patients with severe conditions would be excluded from this study. The selection was based on consecutive sampling and voluntary participation.”
|
||
|
Comments 2: While the discussion mentions the study's limitations, expanding on how these limitations might affect the interpretation of the results would be beneficial. For example, how might the relatively small sample sizes for some patient groups influence the findings?
|
||
|
Response 2: Agree. We have, accordingly, made the necessary revision by emphasizing in the conclusion of this manuscript that a major limitation of this study is the inability to measure the true impact of HRQOL among chronic patients with severe conditions. Although the sample size is small, the results still demonstrated statistically significant findings, showing that the HRQ-6D is sensitive in differentiating between individuals with and without chronic diseases. However, future studies can apply the HRQ-6D to a broader range of populations, including patients with various diseases, and with a larger sample size in order to get a better representation of HRQOL in these populations.
|
||
|
4. Response to Comments on the Quality of English Language |
||
|
Point 1: No comment. |
||
|
|
||
|
5. Additional clarifications |
||
|
None. |
||
